# Alcohol Use and Misuse Among School-Going Adolescents in Thailand: Results of a National Survey in 2015

**DOI:** 10.3390/ijerph16111898

**Published:** 2019-05-29

**Authors:** Supa Pengpid, Karl Peltzer

**Affiliations:** 1ASEAN Institute for Health Development, Mahidol University, Salaya, Phutthamonthon, Nakhon Pathom 73170, Thailand; supaprom@yahoo.com; 2Deputy Vice Chancellor Research and Innovation Office, North West University, Potchefstroom 2520, South Africa

**Keywords:** Alcohol use, alcohol misuse, adolescents, Thailand

## Abstract

The aim of this study was to assess the prevalence of alcohol use and misuse, and to identify its associated factors among in-school adolescents in the 2015 Thailand Global School-based Student Health Survey (GSHS). The sample included 5994 school-going adolescents (mean age 14.5 years, SD = 1.7) from Thailand that responded to the 2015 GSHS. Overall, 22.2% were current alcohol users, 24.3% had ever been drunk, 12.1% had drunk two or more alcoholic drinks in a day in the past 30 days and 10.8% had gotten into trouble because of drinking alcohol. In adjusted Poisson regression analysis, older age, psychological distress, current tobacco use, the consumption of one or more soft drinks a day, school truancy, having been in a physical fight in the past 12 months, and having been seriously injured in the past 12 months were associated with current alcohol use. Older age, psychological distress, current tobacco use and injury also increased the odds for lifetime drunkenness, having two or more drinks in a day and trouble resulting from drinking. Soft drink consumption and having been in a physical fight also increased the odds for lifetime drunkenness and having two or more drinks in a day and school truancy also increased the odds for lifetime drunkenness and trouble resulting from drinking. In addition. Parental tobacco use was associated with lifetime drunkenness and trouble resulting from drinking, cannabis use with trouble resulting from drinking, and parental support was protective from trouble resulting from drinking. There were no significant sex differences regarding any of the four alcohol use indicators. More than one in five school-going adolescents in Thailand use and misuse alcohol, and strategies to prevent alcohol misuse, including a cluster of risk behaviours, are needed.

## 1. Introduction

Globally, alcohol use is a major contributor to morbidity and mortality [1], and is on the increase among young people (10–24 years) [2]. The use of alcohol in young people, in particular during the period of adolescence, significantly increases the risk for developing alcohol use problems in adult life [3]. In a 2007 national household survey in Thailand the current use of alcohol was 17.9% among male and 7.3% among female adolescents (12–19 years old), and the prevalence of past month binge drinking was 9.5% and 3.7% for male and female adolescents, respectively, and drinking until intoxication was 17.3% in males and 7.2% in females [4]. In the 2008 “Thailand Global School-based Student Health Survey (GSHS)”, the prevalence of current alcohol use was 14.8% (21.2% among males and 9.3% among females), and ever been drunk was 19.0% (24.7% among males and 13.6% among females) (12–15 years old) [5]. Among school-going adolescents (13–16 years) in eight “Association of Southeast Asian Nations (ASEAN)”, the prevalence of current alcohol use was high in the Philippines, Thailand and Vietnam (16%–24%), compared to the remaining five countries (1.4%–8.2%) [6]. In the 2013 GSHS in Cambodia, 10.0% of the school students (mean age 15.7 years; SD = 1.8) reported current alcohol use, 10.8% lifetime drunkenness, and 2.8% had problems as a result of drinking [7]. There have been no more recent national studies among school-going adolescents on the prevalence of alcohol use and its correlates in Thailand since 2008. Surveillance of the alcohol use pattern over time is needed to guide health policy in Thailand [8].

Risk factors for alcohol use and misuse among adolescents, may include male gender [4,5,7,9], older age [4,5,7], psychological distress [7,10,11,12,13,14], tobacco use [5,7,11,12], illicit drug use [5,7,11,12], school truancy [12,15], frequent consumption of sugary drinks [16], a lack of peer support [11,13,14], parental substance use [11,17,18] and a lack of parental support [5,7]. In addition, adverse life events, such as experiencing hunger [19], bullying victimization [7,17], fighting or aggressive behaviour [5,11], and injury [5,20] have been found associated with alcohol use and misuse among adolescents.

The aim of this study was to assess the prevalence of alcohol use and misuse and identify its associated factors among in-school adolescents in the 2015 GSHS in Thailand.

## 2. Methods

### 2.1. Study Design and Procedures

The study included a secondary analysis of the 2015 Thailand GSHS, which “employed a two-stage cluster sample design to produce a representative sample of students between the ages of 13 and 17 years in grades 7–12. The first-stage sampling frame consisted of all public and private schools offering education from grades 7 to 12. The second stage of sampling consisted of randomly selecting intact classrooms (using a random start) from each school. All students in the sampled classrooms were eligible to participate in the GSHS” [21]. The study protocol was approved by a Health Research Ethics Committee in Thailand, and “verbal or written consent was obtained from all adolescents and their parents or guardians” [21].

### 2.2. Measures

The GSHS questionnaire was used in this study, consisting of modules on “demographics, alcohol use, tobacco use, drug use, violence, mental health, injury and a range of other health related behaviours, and parental or guardian support” [21]. The questionnaire items used in this paper are described in Table 1. The psychological distress items (no close friends, loneliness, anxiety, suicidal ideation and suicide attempt) were summed, and grouped into 0 = 0 low, 1 = 1 medium and 2–5 = 2 high. The four items on parental or guardian support (supervision, connectedness, bonding and respect for privacy) were summed, and classified into three groups, 0–1 = low, 2 = medium and 3–4 = high support.

### 2.3. Data Analysis

Descriptive statistics were used to present the sample and other characteristics of the study variables. Poisson regression was applied to estimate associations between independent variables (age, sex, psychological distress, current tobacco use, current cannabis use, daily soft drink consumption, school truancy, peer support, parental tobacco use, secondary smoke, parental support, hunger, bullied, in a physical fight, physically attacked and injury) and alcohol use indicators (past month use, ever drunk, two or more drinks in a day and trouble from drinking) by calculating prevalence ratios (PR). Independent variables found significant (*p* < 0.05) in relation to the outcome variable in bivariate analysis were included in the final multivariable model. Both, the 95% confidence intervals and *p*-values were adjusted for the complex study design of the study. *p* < 0.05 was considered significant. Missing data were excluded from the analyses. All analyses were conducted with STATA software version 15.0 (Stata Corporation, College Station, TX, USA).

## 3. Results

Table 2 shows the sample characteristics. Overall, 22.2% were current alcohol users (27.0% among males and 17.9% females), 24.3% had ever been drunk (27.0% among males and 21.8% among females), 12.1% drank two or more alcoholic drinks in a day in the past 30 days (13.8% among males and 10.5% among females) and 10.8% had gotten into trouble as a result of drinking alcohol (14.2% among males and 7.9% among females) (see Table 2).

### Associations with Alcohol Use Indicators

In multivariable Poisson regression analysis, older age (adjusted prevalence ratio = APR: 2.51, 95% Confidence Interval = CI, 1.85–3.40), psychological distress (APR: 1.64, 95% CI: 1.28–2.10), current tobacco use (APR: 2.13; 95% CI: 1.65–2.75), the consumption of one or more soft drinks a day (APR = 1.44, 95% CI: 1.19–1.75), school truancy (APR: 1.26, 95% CI, 1.02–1.56), having been in a physical fight in the past 12 months (APR: 1.26, 95% CI: 1.09–1.58), and having been seriously injured in the past 12 months (APR: 1.43, 95% CI, 1.14–1.79) were associated with current alcohol use. 

Older age, psychological distress, current tobacco use and injury also increased the odds for lifetime drunkenness, having two or more drinks in a day and trouble resulting from drinking. Soft drink consumption and having been in a physical fight also increased the odds for lifetime drunkenness and having two or more drinks in a day, and school truancy also increased the odds for lifetime drunkenness and trouble resulting from drinking. There were no significant sex differences regarding any of the four alcohol use indicators. In addition, parental tobacco use was associated with lifetime drunkenness and trouble resulting from drinking, cannabis use with trouble resulting from drinking, and parental support was protective from trouble resulting from drinking (see Table 3, Table 4, Table 5 and Table 6).

## 4. Discussion

The study aimed to assess the prevalence of alcohol use and misuse and identify its associated factors among in-school adolescents in the 2015 GSHS in Thailand. More than one in five school-going adolescents in Thailand use and misuse alcohol. Compared to the 2008 GSHS in Thailand [5], current alcohol use increased from 14.8% to 22.2% in the 2015 GSHS, in particular among females the prevalence of current alcohol use almost doubled, from 9.3% in 2008 to 17.9% in 2015. Similarly, the prevalence of having ever been drunk increased from 19.0% in the 2008 GSHS to 24.3% in the 2015 GSHS in Thailand [5], with a larger increase in females, from 13.6% to 21.8%, than males, from 24.7% to 27.0%. Among ASEAN countries, the prevalence of current alcohol use was in Thailand the second highest, behind the Philippines (23.7%) [6]. The found past-month prevalence of 22.2% alcohol use in Thailand is similar to the overall past-month alcohol use prevalence among adolescents (12–15 years) from 57 low- and middle-income countries (25.0%) [8]. The prevalence of lifetime drunkenness (24.3%) was in this study higher than in the multi-country study (17.9%) [8], and the prevalence of drinking-related problems (10.8%) was in this study similar to the multi-country prevalence (10.6%) [8]. The increase of alcohol use and misuse among adolescents in Thailand over time calls for increased efforts for strategies to increase abstinence from alcohol and a reduction of alcohol misuse in this population group.

While previous and older studies in Thailand and in the ASEAN region showed a preponderance of alcohol use and misuse among male adolescents [4,5,7,9], this study did not find significant sex differences. It seems that social and cultural norms that hinder females to engage in alcohol use [22] have changed, as they seem to engage in similar ways in alcohol use and misuse as males do, reflecting “increasingly equal rights and conditions between males and females” [8]. It is possible that alcohol-marketing strategies reinforce this trend [8]. Consistent with previous studies [4,5,7,17,18], this study found that alcohol use and misuse increased with age. In fact the legal drinking age was increased from 18 to 20 years in 2008 in Thailand [23], but this seems not to have deterred the adolescents (11–18 years) in this sample to engage in alcohol use and misuse. Older adolescents may have more contact with their peers than younger adolescents, and peer influence may increase alcohol use and misuse [9]. In this study about half of current alcohol users (48.3%) bought themselves alcohol from the store, while 41.0% obtained alcohol from friends, family or gave someone money to buy alcohol (analysis not shown). In Thailand, the alcohol beverage control act bans the sale of alcoholic beverages to persons under the age of 20 years [24]. It is possible that there is poor enforcement of the policy of alcohol sale to minors in Thailand [25]. In a study among Thai youth (15–24 years) it was found that “increases in taxation may prevent drinking initiation and may reduce the harms caused by alcohol” [26].

Consistent with previous studies [5,7,10,11,12,13,14,15,16,20], this survey found that psychological distress, tobacco use, cannabis use, soft drink consumption, school truancy, in a physical fight and injury were associated with alcohol use and/or misuse. Psychological distress that was found to be related to alcohol use in this study may be consistent with a stress coping model, whereby students use alcohol as a means of coping with psychological distress [27,28]. Previous studies have highlighted the clustering of multiple risk behaviours [16,29,30,31], in particular alcohol and tobacco use, but also illicit drug use (such as cannabis use), truancy, interpersonal violence and frequent soft drink (sugar) consumption. The combination of multiple health risk behaviours found in this study reaffirms the problem behaviour theory in vulnerable adolescents [32]. “Problem behaviour theory defines risk behaviour as anything that can interfere with successful psychosocial development and problem behaviour as risk behaviours that elicit either formal or informal social responses designed to control them” [31]. “These may cluster to form a ‘risk behaviour syndrome’ when they serve a common social or psychological developmental function (e.g., affirming individuation from parents, helping to achieve adult status, gaining acceptance from peers), and may help the adolescent to cope with failure, boredom, social anxiety, unhappiness, rejection, social isolation, low self-esteem, or a lack of self-efficacy” [31].

In agreement with some previous studies [11,17,18], this study found that parental tobacco use increased the odds of having trouble from drinking. It is possible that parental substance use serves as a role model for their children to engage also in substance use, such as alcohol [33]. Previous studies [5,7] found that parental support was protective from alcohol use, while this study only found that parental support was protective from trouble as a result from drinking. Contrary to previous investigations [7,17], this study did not find an association between bullying victimization and alcohol use. We found a negative association between bullying victimization and trouble from drinking. This result be may be explained by the possible more passive behaviour of students who are bullied and thus, not engaging in more externalized behaviour getting them into trouble from drinking.

### Study Strength and Limitation

The strength of the study was that a nationally representative sample of adolescents attending public and private schools in Thailand was utilized. The study was cross-sectional, so no causative conclusions can be drawn. Since the sample only included school-going adolescents, no information can be provided on non-school-going adolescents. As the anonymous information collected in the survey was by self-report, responses may have been biased leading to especially underreporting of substance, including alcohol, use. The GSHS questionnaire measures different concepts only with single items, and future studies may include multiple item measures. Certain variables, such as parental and peer alcohol use, were not assessed, and should be included in future studies. The different study variables were assessed using different reference periods, e.g., for alcohol use in the past month and for psychological distress variables in the past year, which should be considered in the interpretation of the results.

## 5. Conclusions

More than one in five school-going adolescents in Thailand use and misuse alcohol. Older age, psychological distress, current tobacco use and injury increased the odds for current alcohol use, lifetime drunkenness, having two or more drinks in a day and trouble resulting from drinking. Soft drink consumption and having been in a physical fight increased the odds for current alcohol use, lifetime drunkenness and having two or more drinks in a day, and school truancy increased the odds for current alcohol use, lifetime drunkenness and trouble resulting from drinking. Parental tobacco use was associated with lifetime drunkenness and trouble resulting from drinking, cannabis use with trouble resulting from drinking, and parental support was protective from trouble resulting from drinking. Efforts to prevent and control alcohol use and misuse should include the clustering of risk behaviours, such as psychological distress, tobacco use, cannabis use, school truancy, physical fighting and being injured.

## Figures and Tables

**Table 1 ijerph-16-01898-t001:** Variable description.

Variables	Question	Response Options (Coding Scheme)
Age	“How old are you?”	11 years old or younger to 18 years old or older
Sex	“What is your sex?”	Male, Female
Alcohol use	“During the past 30 days, on how many days did you have at least one drink containing alcohol?”	1 = 0 days to 7 = All 30 days (coded 1 = 0 and 2–7 = 1)
	“During the past 30 days, on the days you drink alcohol, how many drinks did you usually drink per day?”	1 = Not drink in the past 30 days to 7 = 5 or more drinks (coded 1–3 = 0 and 4–7 = 1)
	“During your life, how many times did you drink so much alcohol that you were really drunk?”	1 = 0 times and 4 = 10 or more times (coded 1 = 0 and 2–4 = 1)
	“During your life, how many times have you got into trouble with your family or friends, missed school, or got into fights, as a result of drinking alcohol?”	1 = 0 times and 4 = 10 or more times (coded 1 = 0 and 2–4 = 1)
No close friends	“How many close friends do you have?”	1 = 0 to 4 = 3 or more (coded 1+ = 0, 0 = 1)
Anxiety	“During the past 12 months, how often have you been so worried about something that you could not sleep at night?”	1 = never to 5 = always (coded 1–3 = 0 and 4–5 = 1)
Loneliness	“During the past 12 months, how often have you felt lonely?”	1 = never to 5 = always (coded 1–3 = 0 and 4–5 = 1)
Suicide ideation	“During the past 12 months, did you ever seriously consider attempting suicide?”	Yes, No
Suicide attempt	“During the past 12 months, how many times did you actually attempt suicide?”	1 = 0 times to 5 = 6 or more times (coded 1 = 0 and 2–5 = 1)
Current tobacco use	“During the past 30 days, on how many days did you smoke cigarettes/use any tobacco products other than cigarettes, such as cigars, *haraku*, or electronic cigarettes?”	1 = 0 days to 7 = All 30 days (coded 1 = 0 and 2–7 = 1)
Cannabis use	“During the past 30 days, how many times have you used marijuana (also called *nua*)?”	1 = 0 days to 7 = All 30 days (coded 1 = 0 and 2–5 = 1)
Soft drinks	“During the past 30 days, how many times per day did you usually drink carbonated soft drinks, such as *Mam-ud-lon*?”	1 = not in the past days to 7 = 5 or more times per day (coded 1–2 = 0 and 3–7 = 1)
School truancy	“During the past 30 days, on how many days did you miss classes or school without permission?”	1 = 0 days to 5 = 10 or more days (coded 1 = 0 and 2–5 = 1)
Peer support	“During the past 30 days, how often were most of the students in your school kind and helpful?”	1 = never to 5 = always (coded 1–3 = 0 and 4–5 = 1)
Either or both parents use tobacco	“Which of your parents or guardians use any form of tobacco?”	1 = neither, 2 = my father or male guardian, 3 = my mother or female guardian, 4 = both
Secondary smoke	“During the past 7 days, on how many days have people smoked in your presence?“	1 = 0 days to 5 = All seven days (coded 1 = 0 and 2–5 = 1)
Parental supervision	“During the past 30 days, how often did your parents or guardians check to see if your homework was done?”	1 = never to 5 = always (coded 1–3 = 0 and 4–5 = 1)
Parental connectedness	“During the past 30 days, how often did your parents or guardians understand your problems and worries?”	1 = never to 5 = always (coded 1–3 = 0 and 4–5 = 1)
Parental bonding	“During the past 30 days, how often did your parents or guardians really know what you were doing with your free time?”	1 = never to 5 = always (coded 1–3 = 0 and 4–5 = 1)
Parental respect for privacy	“During the past 30 days, how often did your parents or guardians go through your things without your approval?”	1 = never to 5 = always (coded 1–3 = 0 and 4–5 = 1)
Hunger	“During the past 30 days, how often did you go hungry because there was not enough food in your home?"	1 = never to 5 = always (coded 1–3 = 0 and 4–5 = 1)
Bullied	“During the past 30 days, on how many days were you bullied?”	1 = 0 days to 7 = All 30 days (coded 1 = 0 and 2–7 = 1)
In a physical fight	“During the past 12 months, how many times were you in a physical fight?”	1 = 0 times to 8 = 12 or more times (coded 1 = 0 and 2–8 = 1)
Physically attacked	“During the past 12 months, how many times were you physically attacked?”	1 = 0 times to 8 = 12 or more times (coded 1 = 0 and 2–8 = 1)
Injury	“During the past 12 months, how many times were you seriously injured?”	1 = 0 times to 8 = 12 or more times (coded 1 = 0 and 2–8 = 1)

**Table 2 ijerph-16-01898-t002:** Sample characteristics of adolescents in Thailand, GSHS, 2015 (*N* = 5994).

Variable (#Missing Cases)	*N* (%)
Age (years) (#17)	
≤13	1951 (30.7)
14	1216 (21.1)
15	998 (17.2)
≥16	1712 (31.1)
Sex (#31)	
Female	3335 (52.9)
Male	2528 (47.1)
Psychological distress (#399)	
0	4010 (73.4)
1	822 (14.6)
2 or more	663 (12.0)
Current tobacco use (#114)	800 (14.1)
Current cannabis use (#176)	275 (5.1)
Soft drinks (1 or more/day) (#25)	3103 (56.1)
School truancy (#132)	1193 (20.4)
Peer support (#141)	2194 (39.5)
Parental tobacco use (#59)	1733 (31.8)
Secondary smoke (#117)	2302 (41.6)
Parental support (#409)	
0–1	2721 (49.5)
2	1488 (27.8)
3–4	1276 (22.7)
Hunger (#12)	215 (3.9)
Bullied (#664)	1621 (29.3)
In physical fight (#20)	1585 (25.7)
Physically attacked (#141)	1545 (26.2)
Injury (#1071)	2038 (39.6)
Alcohol use indicators	
Current alcohol use (#190)	1153 (22.2)
Ever drunk (#257)	1154 (24.3)
Two or more drinks (#152)	614 (12.1)
Trouble from drinking (#314)	526 (10.8)

**Table 3 ijerph-16-01898-t003:** Associations with current alcohol use.

Variable	Current Alcohol Use	Not Current Alcohol Use	Current Alcohol Use	
	*N* (%)	*N* (%)	CPR (95% CI)	APR (95% CI)
Age (years)				
≤13	261 (14.4)	1601 (85.6)	1 (Reference)	1 (Reference)
14	233 (18.6)	945 (81.4)	1.30 (0.99–1.70)	1.59 (1.15–2.21) **
15	200 (21.9)	767 (78.1)	1.52 (1.13–2.96) **	1.57 (1.11–2.23) *
≥16	457 (32.4)	1223 (67.6)	2.26 (1.68–3.04) ***	2.51 (1.85–3.40) ***
Sex				
Female	541 (17.9)	2724 (82.1)	1 (Reference)	1 (Reference)
Male	604 (27.0)	1805 (73.0)	1.51 (1.26–1.80) ***	1.04 (0.83–1.31)
Psychological distress				
0	619 (16.8)	3328 (83.2)	1 (Reference)	1 (Reference)
1	186 (25.2)	610 (74.8)	1.50 (1.31–1.72) ***	1.29 (1.02–1.64) *
2 or more	215 (39.3)	413 (60.7)	2.34 (1.93–2.83) ***	1.64 (1.28–2.10) ***
Current tobacco use	456 (65.2)	278 (34.8)	4.18 (3.73–4.70) ***	2.13 (1.65–2.75) ***
Current cannabis use	107 (71.5)	75 (28.5)	3.76 (3.22–4.39) ***	0.91 (0.69–1.21)
Soft drink consumption	756 (26.7)	2234 (73.3)	1.62 (1.41–1.86) ***	1.44 (1.19–1.75) ***
School truancy	395 (41.5)	721 (58.5)	2.44 (2.12–2.80) ***	1.26 (1.02–1.56) *
Peer support	365 (18.5)	1784 (81.5)	0.76 (0.63–0.93) **	0.90 (0.72–1.14)
Parental tobacco use	419 (26.0)	1252 (74.0)	1.31 (1.11–1.54) ***	1.06 (0.86–1.31)
Secondary smoke	654 (31.0)	1561 (69.0)	2.05 (1.83–2.31) ***	1.20 (0.95–1.52)
Parental support				
0–1	622 (25.3)	1997 (83.2)	1 (Reference)	1 (Reference)
2	274 (21.0)	1183 (74.8)	0.83 (0.69–0.99) *	0.97 (0.80–1.16)
3–4	153 (13.1)	1100 (60.7)	0.52 (0.38–0.70) ***	0.87 (0.64–1.18)
Hunger	50 (29.7)	148 (70.3)	1.36 (0.96–1.93)	-
Bullied	438 (32.0)	1099 (78.0)	1.89 (1.57–2.27) ***	1.10 (0.87–1.38)
In physical fight	512 (37.4)	985 (62.6)	2.19 (1.89–2.54) ***	1.26 (1.01–1.58) *
Physically attacked	444 (32.6)	1018 (67.4)	1.77 (1.44–2.17) ***	1.05 (0.77–1.43)
Injury	589 (34.0)	1351 (66.0)	2.35 (1.96–2.83) ***	1.43 (1.14–1.79) **

CR = crude prevalence ratio; AR = adjusted prevalence ratio; CI = confidence interval; *** *p* < 0.001; ** *p* < 0.01; * *p* < 0.05.

**Table 4 ijerph-16-01898-t004:** Associations with ever having been drunk.

Variable	Ever Drunk	Never Drunk	Ever Drunk	
	*N* (%)	*N* (%)	CPR (95% CI)	APR (95% CI)
Age (years)				
≤13	212 (14.2)	1632 (85.8)	1 (Reference)	1 (Reference)
14	210 (17.2)	949 (82.8)	1.21 (0.91–1.60)	1.29 (0.97–1.70)
15	199 (23.1)	758 (76.9)	1.63 (1.23–2.16) ***	1.62 (1.14–2.30) **
≥16	531 (39.4)	1130 (60.6)	2.77 (2.09–3.69) ***	3.33 (2.49–4.46) ***
Sex				
Female	572 (21.8)	2670 (78.2)	1 (Reference)	1 (Reference)
Male	573 (27.0)	1793 (73.0)	1.24 (1.06–1.44) **	0.92 (0.78–1.08)
Psychological distress				
0	646 (19.9)	3270 (80.1)	1 (Reference)	1 (Reference)
1	198 (29.3)	592 (70.7)	1.48 (1.25–1.74) ***	1.19 (0.97–1.47)
2 or more	217 (40.6)	401 (59.4)	2.04 (1.71–2.43) ***	1.42 (1.14–1.78) **
Current tobacco use	404 (61.9)	293 (38.1)	3.33 (2.89–3.84) ***	2.01 (1.64–2.46) ***
Current cannabis use	149 (63.3)	71 (36.7)	2.92 (2.46–3.47) ***	1.09 (0.86–1.39)
Soft drink consumption	725 (28.4)	2210 (71.6)	1.50 (1.30–1.72) ***	1.31 (1.10–1.56) **
School truancy	360 (40.3)	736 (59.7)	2.00 (1.74–2.28) ***	1.37 (1.13–1.65) ***
Peer support	401 (23.0)	1719 (77.0)	0.92 (0.78–1.09)	-
Parental tobacco use	439 (301.)	1216 (69.9)	1.42 (1.25–1.65) ***	1.21 (1.01–1.45) *
Secondary smoke	628 (32.9)	1553 (67.1)	1.86 (1.65–2.09) ***	1.13 (0.94–1.36)
Parental support				
0–1	602 (27.4)	1990 (72.6)	1 (Reference)	1 (Reference)
2	283 (23.7)	1163 (76.3)	0.87 (0.72–1.03)	1.02 (0.84–1.25)
3–4	181 (17.2)	1068 (82.8)	0.63 (0.48–0.82) ***	0.85 (0.63–1.13)
Hunger	56 (33.8)	137 (66.2)	1.42 (1.06–1.89) *	0.99 (0.61–1.64)
Bullied	373 (28.8)	1128 (71.2)	1.41 (1.20–1.66) ***	0.83 (0.69–1.01)
In physical fight	475 (37.3)	991 (62.7)	1.87 (1.65–2.13) ***	1.34 (1.14–1.57) ***
Physically attacked	396 (30.8)	1039 (69.2)	1.41 (1.21–1.64)	-
Injury	560 (34.7)	1337 (65.3)	2.05 (1.77–2.37) ***	1.53 (1.32–1.78) ***

CR = crude prevalence ratio; AR = adjusted prevalence ratio; CI = confidence interval; *** *p* < 0.001; ** *p* < 0.01; * *p* < 0.05.

**Table 5 ijerph-16-01898-t005:** Associations with ≥2 drinks/day.

Variable	≥2 Drinks/Day	<2 or No Drinks/Day	≥2 Drinks/Day	
	*N* (%)	*N* (%)	CPR (95% CI)	APR (95% CI)
Age (years)				
≤13	116 (6.3)	1774 (93.7)	1 (Reference)	1 (Reference)
14	100 (7.5)	1084 (92.5)	1.20 (0.85–1.69)	1.13 (0.81–1.59)
15	109 (12.7)	860 (87.3)	2.03 (1.34–3.08) ***	1.55 (1.06–2.26) *
≥16	288 (20.4)	1394 (79.6)	3.25 (2.21–4.77) ***	3.20 (2.28–4.51) ***
Sex				
Female	297 (10.5)	2982 (89.5)	1 (Reference)	1 (Reference)
Male	311 (13.8)	2121 (86.2)	1.32 (1.02–1.71) *	0.89 (0.63–1.26)
Psychological distress				
0	317 (8.8)	3632 (91.2)	1 (Reference)	1 (Reference)
1	95 (13.2)	707 (86.8)	1.50 (1.16–1.95) **	1.11 (0.81–1.53)
2 or more	142 (25.6)	502 (74.4)	2.90 (2.32–3.63) ***	1.78 (1.37–2.31) ***
Current tobacco use	256 (35.5)	495 (64.5)	4.19 (3.31–5.29) ***	2.92 (2.07–4.14) ***
Current cannabis use	89 (38.4)	162 (61.6)	3.67 (2.88–4.68) ***	1.00 (0.67–1.47)
Soft drink consumption	404 (14.6)	2609 (85.4)	1.65 (1.28–2.14) ***	1.51 (1.07–2.14) *
School truancy	212 (22.4)	930 (66.6)	2.41 (1.94–3.00) ***	1.29 (0.97–1.73)
Peer support	203 (10.8)	1951 (89.2)	0.84 (0.65–1.08)	-
Parental tobacco use	214 (14.0)	1479 (86.0)	1.19 (1.00–1.65) *	0.97 (0.73–1.31)
Secondary smoke	372 (17.5)	1867 (82.5)	2.21 (1.78–2.73) ***	1.17 (0.90–1.52)
Parental support				
0–1	325 (13.4)	2323 (86.6)	1 (Reference)	1 (Reference)
2	147 (11.2)	1321 (88.8)	0.84 (0.67–1.05)	1.03 (0.80–1.33)
3–4	90 (8.0)	1169 (92.0)	0.60 (0.44–0.81) ***	1.01 (0.69–1.46)
Hunger	29 (16.5)	178 (83.5)	1.39 (0.87–2.21)	-
Bullied	228 (17.0)	1337 (83.0)	1.88 (1.42–2.49) ***	1.16 (0.82–1.64)
In physical fight	290 (20.5)	1241 (79.5)	2.25 (1.89–2.67) ***	1.55 (1.16–2.07) **
Physically attacked	237 (16.8)	1252 (83.2)	1.61 (1.26–2.06) ***	0.85 (0.64–1.14)
Injury	323 (19.5)	1637 (80.5)	2.70 (2.10–3.48) ***	1.52 (1.08–2.14) *

CR = crude prevalence ratio; AR = adjusted prevalence ratio; CI = confidence interval; *** *p* < 0.001; ** *p* < 0.01; * *p* < 0.05.

**Table 6 ijerph-16-01898-t006:** Associations with trouble from drinking.

Variable	Trouble from Drinking	No Trouble from Drinking	Trouble from Drinking	
	*N* (%)	*N* (%)	CPR (95% CI)	APR (95% CI)
Age (years)				
≤13	111 (6.8)	1707 (93.2)	1 (Reference)	1 (Reference)
14	102 (8.8)	1048 (91.2)	1.30 (0.85–2.01)	1.83 (1.10–3.03) *
15	81 (8.7)	863 (91.3)	1.29 (0.77–2.13)	0.72 (0.33–1.57)
≥16	231 (17.2)	1422 (82.8)	2.53 (1.56–4.13) ***	3.01 (1.88–4.85) ***
Sex				
Female	216 (7.9)	3001 (92.1)	1 (Reference)	1 (Reference)
Male	305 (14.2)	2031 (85.8)	1.81 (1.40–2.32) ***	0.81 (0.53–1.25)
Psychological distress				
0	214 (6.6)	3681 (93.4)	1 (Reference)	1 (Reference)
1	88 (12.5)	688 (87.5)	1.90 (1.45–2.48) ***	1.78 (1.18–2.69) **
2 or more	136 (27.0)	460 (73.0)	4.08 (3.38–4.94) ***	2.53 (1.85–3.45) ***
Current tobacco use	274 (43.0)	410 (57.0)	7.21 (5.57–9.33) ***	3.37 (2.03–5.60) ***
Current cannabis use	126 (61.7)	91 (38.3)	7.64 (6.17–9.46) ***	2.43 (1.73–3.41) ***
Soft drink consumption	344 (13.2)	2557 (86.8)	1.70 (1.36–2.12) ***	1.35 (0.92–1.97)
School truancy	229 (26.3)	854 (73.7)	3.84 (3.30–4.48) ***	1.57 (1.17–2.11) **
Peer support	152 (8.6)	1964 (91.4)	0.70 (0.52–0.95) *	1.21 (0.80–1.83)
Parental tobacco use	231 (15.4)	1402 (84.6)	1.84 (1.46–2.31) ***	1.70 (1.28–2.26) ***
Secondary smoke	318 (15.8)	1846 (84.2)	2.32 (1.93–2.79) ***	0.89 (0.60–1.34)
Parental support				
0–1	295 (13.0)	2275 (87.0)	1 (Reference)	1 (Reference)
2	125 (9.9)	1306 (90.1)	0.77 (0.58–1.02)	0.75 (0.53–1.06)
3–4	53 (4.8)	1190 (95.2)	0.37 (0.23–0.59) ***	0.58 (0.36–0.95) *
Hunger	27 (17.2)	161 (82.8)	1.63 (1.01–2.63) *	0.45 (0.15–1.32)
Bullied	211 (16.2)	1255 (83.8)	2.12 (1.64–2.73) ***	0.58 (0.42–0.79) ***
In physical fight	266 (20.8)	1183 (79.2)	2.72 (2.23–3.33) ***	1.24 (0.87–1.75)
Physically attacked	232 (17.9)	1175 (82.1)	2.13 (1.69–2.67) ***	1.44 (0.99–2.08)
Injury	304 (18.1)	1563 (81.9)	3.31 (2.51–4.36) ***	1.99 (1.36–2.89) ***

CR = crude prevalence ratio; AR = adjusted prevalence ratio; CI = confidence interval; *** *p* < 0.001; ** *p* < 0.01; * *p* < 0.05.

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
