# Peer review of "Alcohol Use and Misuse Among School-Going Adolescents in Thailand: Results of a National Survey in 2015"

_ijerph, 2019, doi:10.3390/ijerph16111898_

Round 1

Reviewer 1 Report

o  In the introduction, the research gaps in were not captured. Some research has been done on alcohol use in school going children but the Authors didn’t bring out the research gaps to justify the study objectives.

o  In the introduction, the authors mentioned that they employed the social ecological model (SEM). There are five nested, hierarchical levels of the SEM: Individual, interpersonal, community, organizational, and policy/enabling environment. However, the focus of the authors left out community, organizational, and policy/enabling environment factors.

o  Under methods, section 2.1 is titled Sample and procedures but there is no mention of the sample. The title for section 2.1 should have been Study design and procedures.

o  The sample size for the students included in the study was not provided in the methods section, though it appears later in the results.

o  Data analysis: The dependent and independent variables were not defined and therefore the analysis section is difficult to comprehend. The authors should have defined the dependent and independent variables, and explained the measurement criteria for each of the variables.

o  What was the justification of using odds ratios in the analysis?

o  In the results section, Table 3 summarizes 4 outcomes variables including: Current alcohol use; Ever drunk; Two or more drinks; and Trouble from drinking. However, the table leaves out a lot of useful information specifically on the proportions. For example considering current alcohol use as the outcome, for the different independent variables, the table doesn’t provide the proportions of those that were using alcohol currently and those that were not using alcohol. Without providing the proportions, it isn’t possible to verify if the odds ratios provided are valid. It would have been appropriate to present the different outcome variables in different tables with details on proportions, the unadjusted odds ratios and the adjusted odds ratios.

o  In the discussion section, the first sentence is not grammatically correct. “The aimed to assess…….”

o  The discussion is shallow. It would have been appropriate for the authors to introduce the discussion section by highlighting the major research findings and then follow that up by explaining reasons why results appear the way they appear as well as comparing results to existing literature. In the discussion section, authors didn’t give explanations for the results. For example why did parental tobacco use increase the odds of having trouble from drinking? What could be the reasons for that result?

Author Response

In the introduction, the research gaps in were not captured. Some research has been done on alcohol use in school going children but the Authors didn’t bring out the research gaps to justify the study objectives.

Response: more is added

In the introduction, the authors mentioned that they employed the social ecological model (SEM). There are five nested, hierarchical levels of the SEM: Individual, interpersonal, community, organizational, and policy/enabling environment. However, the focus of the authors left out community, organizational, and policy/enabling environment factors.

Response: The SEM is removed

Under methods, section 2.1 is titled Sample and procedures but there is no mention of the sample. The title for section 2.1 should have been Study design and procedures.

Response: this is changed to “Study design and procedures)

o  The sample size for the students included in the study was not provided in the methods section, though it appears later in the results.

Data analysis: The dependent and independent variables were not defined and therefore the analysis section is difficult to comprehend. The authors should have defined the dependent and independent variables, and explained the measurement criteria for each of the variables.

Response: variables are added, and defined in the measures section

What was the justification of using odds ratios in the analysis?

Response: changed to prevalence ratios

In the results section, Table 3 summarizes 4 outcomes variables including: Current alcohol use; Ever drunk; Two or more drinks; and Trouble from drinking. However, the table leaves out a lot of useful information specifically on the proportions. For example considering current alcohol use as the outcome, for the different independent variables, the table doesn’t provide the proportions of those that were using alcohol currently and those that were not using alcohol. Without providing the proportions, it isn’t possible to verify if the odds ratios provided are valid. It would have been appropriate to present the different outcome variables in different tables with details on proportions, the unadjusted odds ratios and the adjusted odds ratios.

Response: added accordingly

In the discussion section, the first sentence is not grammatically correct. “The aimed to assess…….”

Response: Corrected

The discussion is shallow. It would have been appropriate for the authors to introduce the discussion section by highlighting the major research findings and then follow that up by explaining reasons why results appear the way they appear as well as comparing results to existing literature. In the discussion section, authors didn’t give explanations for the results. For example why did parental tobacco use increase the odds of having trouble from drinking? What could be the reasons for that result?

Response: more is added

Reviewer 2 Report

1.      It would be helpful if the authors provided additional rationale for examining the research questions within the introduction, other than that there is not more recent data in this sample since 2008.  

2.      It is unclear why all independent variables were transformed into binary variables. As this significantly reduces variability in the analyses, a strong rationale should be provided. Further, the distinctions in these scales are very different between measures, making it difficult to compare effects. Most notable is the parental tobacco use variable in which the original scale cannot be considered ordinal, but seems to be treated as such. Additionally, some variables are coded as none (or zero) versus any (greater than 1), while other have more arbitrary cut-offs. If these predictor variables are to remain binary, more rationale for cut-offs is needed.

3.      How were p-values adjusted? Additional detail to the analytic method is needed.

4.      Missingness for the four outcome variables should be clearly presented, and determined not to be predicted by any of the variables of interest. Further, are the percentages presented in Table 2 the percentage of the entire sample or the sample after missing data was removed?

5.      The findings should be placed in additional context of the current state of the literature. For instance, line 148 discussing potential effects of peers on drinking; there is significant literature on this topic that should be reviewed and cited in order to interpret these findings.

6.      Additional background for theories referenced should be provided. For example, in the discussion, lines 144-145, theories such as the problem behavior theory should be defined and discussed with additional detail and in the context of the current findings.

7.      The conclusion paragraph is over-simplified, as individual outcomes and predictors are paired in a single sentence. They should be identified separately in order to prevent inaccurate conclusions.

8.      More detail on how study procedures handled the endorsement of suicidal ideation is warranted.

9.      There should be limited use of subjective language throughout the manuscript (e.g. “alcohol use was high”).

10.  Language regarding gender should be consistent throughout the manuscript (i.e. male v. female or boys v. girls).

11.  Within the introduction, a more precise review of results would help to put this research in context. For example, line 39 reviews research from 2013 GSHS and refers to “school students” but does not provide ages for this group.

12.  The manuscript would benefit from significant copy editing for grammar and spelling.

13.  Different reference styles are used in the manuscript and should be resolved.

Author Response

1.It would be helpful if the authors provided additional rationale for examining the research questions within the introduction, other than that there is not more recent data in this sample since 2008.  

Response: more is added

2.      It is unclear why all independent variables were transformed into binary variables. As this significantly reduces variability in the analyses, a strong rationale should be provided. 

Response: Some variables are changed accordingly.

Further, the distinctions in these scales are very different between measures, making it difficult to compare effects. 

Response: This is added under study limitations

Most notable is the parental tobacco use variable in which the original scale cannot be considered ordinal, but seems to be treated as such. 

Response: not sure what is wrong with this, either any parent or guardian uses tobacco or not

Additionally, some variables are coded as none (or zero) versus any (greater than 1), while other have more arbitrary cut-offs. If these predictor variables are to remain binary, more rationale for cut-offs is needed.

Response: Some variables are changed accordingly.

3.      How were p-values adjusted? Additional detail to the analytic method is needed.

Response: Explained in data analysis section

4.      Missingness for the four outcome variables should be clearly presented, and determined not to be predicted by any of the variables of interest.

Response: added in Table 1

 Further, are the percentages presented in Table 2 the percentage of the entire sample or the sample after missing data was removed?

Response: all percentages are for the whole sample

5.      The findings should be placed in additional context of the current state of the literature. For instance, line 148 discussing potential effects of peers on drinking; there is significant literature on this topic that should be reviewed and cited in order to interpret these findings.

Response: more is added (although the above point was no longer relevant)

6.      Additional background for theories referenced should be provided. For example, in the discussion, lines 144-145, theories such as the problem behavior theory should be defined and discussed with additional detail and in the context of the current findings.

Response: more is added

7.      The conclusion paragraph is over-simplified, as individual outcomes and predictors are paired in a single sentence. They should be identified separately in order to prevent inaccurate conclusions.

Response: Corrected

8.      More detail on how study procedures handled the endorsement of suicidal ideation is warranted.

Response: there is no information available in this

9.      There should be limited use of subjective language throughout the manuscript (e.g. “alcohol use was high”).

Response: Corrected

10.  Language regarding gender should be consistent throughout the manuscript (i.e. male v. female or boys v. girls).

Response: Corrected

11.  Within the introduction, a more precise review of results would help to put this research in context. For example, line 39 reviews research from 2013 GSHS and refers to “school students” but does not provide ages for this group.

Response: added

12.  The manuscript would benefit from significant copy editing for grammar and spelling.

Response: Corrected

13.  Different reference styles are used in the manuscript and should be resolved

Response: Corrected

Many thanks for your comments!

Reviewer 3 Report

The authors first explored the prevalence of alcohol use among school-age Thai adolescents and then explored several socioecological factors behind the usage. 

A foremost question and perhaps also as a concern that came to my mind is whether a study on pre-consent age individuals has cleared all IRB requirements and how was the survey protocol deployed. This point wouldn't be raised if the study focused on adults, but substance use among adolescents has ethical implications. I wish the authors could describe in their article how ethical approval and IRB requirements were fulfilled beyond a simple statement of permission being obtained. . 

I like how the measurement of the variables are concisely described in a table.

There are some theoretical coherence issues in how the socioecological factors are conceptualized in this study. The study has incorporated four groups of variables, classified into personal attributes, parental attributes, and environmental stressors. Yet, which among these factors are social and which are ecological? By ecological, scholars typically refer to some contextual overall forces beyond individual control and unaffected by individual characteristics that originate at a higher ecological unit, such as the GDP of a city, the policy austerity of a jurisdiction. But the environmental stressors described in this study seem to be a misnomer for simply adverse life events and incidental stressors. Similarly, social factors should not be a post-hoc assemblage of what individual causes fail to explain. 

I am generally satisfied with the methodology. 

Author Response

Many thanks for your comments.

The authors first explored the prevalence of alcohol use among school-age Thai adolescents and then explored several socioecological factors behind the usage. 

A foremost question and perhaps also as a concern that came to my mind is whether a study on pre-consent age individuals has cleared all IRB requirements and how was the survey protocol deployed. This point wouldn't be raised if the study focused on adults, but substance use among adolescents has ethical implications. I wish the authors could describe in their article how ethical approval and IRB requirements were fulfilled beyond a simple statement of permission being obtained. . 

Response: as below

The study protocol was approved by a Health Research Ethics Committee in Thailand, and “verbal or written consent was obtained from all adolescents and their parents or guardians. ” [21]

I like how the measurement of the variables are concisely described in a table.

There are some theoretical coherence issues in how the socioecological factors are conceptualized in this study. The study has incorporated four groups of variables, classified into personal attributes, parental attributes, and environmental stressors. Yet, which among these factors are social and which are ecological? By ecological, scholars typically refer to some contextual overall forces beyond individual control and unaffected by individual characteristics that originate at a higher ecological unit, such as the GDP of a city, the policy austerity of a jurisdiction. But the environmental stressors described in this study seem to be a misnomer for simply adverse life events and incidental stressors. Similarly, social factors should not be a post-hoc assemblage of what individual causes fail to explain. 

Response: Since the grouping of variables is problematic, the groupings are removed

I am generally satisfied with the methodology.